# Cardinality-Regularized Hawkes-Granger Model

**Tsuyoshi Idé**
IBM Research, T. J. Watson Research Center
`tide@us.ibm.com`

**Georgios Kollias**
IBM Research, T. J. Watson Research Center
`gkollias@us.ibm.com`

**Dzung T. Phan**
IBM Research, T. J. Watson Research Center
`phandu@us.ibm.com`

**Naoki Abe**
IBM Research, T. J. Watson Research Center
`nabe@us.ibm.com`

## Abstract

We propose a new sparse Granger-causal learning framework for temporal event data. We focus on a specific class of point processes called the Hawkes process. We begin by pointing out that most of the existing sparse causal learning algorithms for the Hawkes process suffer from a singularity in maximum likelihood estimation. As a result, their sparse solutions can appear only as numerical artifacts. In this paper, we propose a mathematically well-defined sparse causal learning framework based on a cardinality-regularized Hawkes process, which remedies the pathological issues of existing approaches. We leverage the proposed algorithm for the task of instance-wise causal event analysis, where sparsity plays a critical role. We validate the proposed framework with two real use-cases, one from the power grid and the other from the cloud data center management domain.

## 1 Introduction

The *Hawkes process* [14] is one of the most popular models for analyzing temporal events in the machine learning (ML) community. It has been applied in a variety of application areas including analysis of human activities on social networks [23, 36, 39, 15, 38, 22, 2, 44], healthcare event analysis [6, 12], search query analysis [21, 20], and even water pipe maintenance [48]. In the studies of Hawkes processes, there have been two major milestones to date. One is the *minorization-maximization (MM)* algorithm [16]. The other is *Granger causal analysis* through Hawkes processes.

The first milestone was marked by Veen and Schoenberg [42]. Based on the intuition of branching process of earthquake aftershocks, they introduced the first MM-based maximum likelihood algorithm, which is often loosely referred to as EM (expectation-maximization) due to their similarity (See Eq. (14) and the discussion below). As their paper pointed out, the standard gradient-based maximum likelihood estimation (MLE) of multivariate Hawkes processes suffered from numerical stability issues, limiting their applicability in practice.

The second milestone was achieved by a few pioneers including Kim et al. [18] who first proposed an approach to Granger causal learning through the Hawkes process; Zhou et al. [51] who introduced $\ell_1$ regularized MLE of a multivariate Hawkes process in the context of Granger causal analysis; and Eichler et al. [9] who theoretically established the equivalence between the Hawkes-based causality and the Granger causality [13].

Given these achievements and the well-known importance of sparsity in Granger causal learning [3, 24], the MM algorithm combined with a sparsity-enforcing regularizer would seem to be a promising path for Granger-causal analysis for stochastic events. This is especially true when the main interest is in analyzing *causal triggering mechanism* of event *instances* since the MM framework provides instance-wise triggering probabilities as a side product. Interestingly, however, the likelihood

35th Conference on Neural Information Processing Systems (NeurIPS 2021).

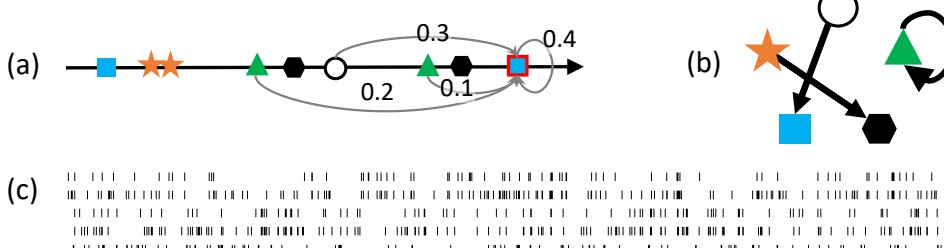

Figure 1: The Hawkes-Granger model allows two different levels of causal analysis: (a) instance-wise and (b) type-wise, in which well-defined *sparsity* is essential for causal diagnosis. (c) Example of five-variate point process data, where each '|' represents an event instance.

function of the MM algorithm has a singularity that in fact prohibits any sparse solutions. Despite its significance, to date little attention has been paid to this issue in the ML community.

In this paper, we provide a mathematically well-defined solution to sparse causal learning on temporal event data for the first time. Specifically, we introduce a novel *cardinality-regularized* MM framework for the Hawkes process. Although cardinality- or $\ell_0$-regularization is known as an ideal way of enforcing sparsity, it makes optimization problems generally hard to solve. Even checking the feasibility is NP-complete [17]. We will show that the Hawkes process in fact has a semi-analytic solution under an $\ell_0$-constraint, which is free from pathological issues due to the logarithmic singularity at zero, unlike existing $\ell_1$-regularization approaches [51, 46]. By leveraging the well-defined sparsity and the side product provided by the MM framework, we tackle the task of instance-wise causal diagnosis in real world domains. Since any event instance occurring in the past is a potential candidate cause of a given event of interest, achieving sufficient sparsity in the learned causal structure has proven critical in these applications. See also Fig. 1 for illustration.

To the best of our knowledge, this work represents the first $\ell_0$-regularized MM learning framework for the Hawkes process applied to simultaneous instance- and type-level causal analysis.

## 2 Related work

Related work relevant to this paper can be categorized into the following five categories.

**Graphical Granger models**: For multivariate temporal data, sparse graphical modeling with a (group) lasso penalty [3, 24, 25] has been a standard approach to Granger causal analysis in the ML community. It replaced subtle independence tests with sparse matrix estimation in a systematic way, and significantly expanded the application field of Granger causal analysis. Their neural extension has also been proposed recently [40]. However, most of those studies use vector autoregressive (VAR) or state-space models (including recurrent neural networks (RNNs)) and thus assume as the input multivariate time-series data with a regular time interval. As a result, they cannot handle event sequences without a certain aggregation/smoothing operation and thus it is not possible to analyze causal structure of event *instances*. We will show that event aggregation can seriously impact significantly impact the accuracy of causal analysis Sec. 6.

**Conventional MLE**: Since Hawkes' original proposal [14], gradient-based MLE has been a mainstream approach to fitting Hawkes processes (see, e.g. [34]). In particular, it has been used extensively in the field of seismology [32]. However, as Veen and Schoenberg [42] and Mark and Weber [26] convincingly demonstrated, gradient-based MLE tends to have numerical instability issues. Unless detailed prior knowledge on the data generating mechanism is available, which is indeed the case in seismology, its applicability to industrial stochastic event data can be limited. Stochastic optimization can be leveraged (e.g. [31]), but it brings in extra variability to inference. In addition to conventional MLE, there are a few other methods to fit multivariate Hawkes processes, such as least squares [47, 5] and moment matching [1]. They trade off the direct linkage to the likelihood principle for computational stability, and are out of our scope. This paper focuses on how to enhance the MM algorithm through cardinality regularization for a sparser and thus more interpretable solution.

**Neural Hawkes models**: Neural point process models typically replace all or part of the terms of the Hawkes intensity function to capture nonlinear temporal dependencies [8, 45, 27, 41, 33, 11, 52]. However, most of the works either assume regular-interval time-series or lack the perspective of instance-wise causal triggering. The transformer architecture [50], on the other hand, allows extracting instance-level information, but the self-attention filter essentially represents auto-correlation rather than Granger causality, making it inapplicable to event-causal diagnosis. Also, the positional encoding approach, originally introduced in a machine translation task, where the notion of time-stamp does not exist, implicitly assumes a regular time grid, partly sharing the same limitation as VAR- and RNN-based methods. Finally, it lacks systematic sparsity-enforcing mechanisms for better interpretability.

**Sparse MM algorithms**: As discussed in Introduction, Zhou et al. [51] and Xu et al. [46] are the pioneers who first attempted to combine sparsity-enforcing regularizers with the MM algorithm. Unfortunately, due to the logarithmic singularity of the likelihood function, sparsity can only be achieved as a numerical artifact under $\ell_1$- and $\ell_{2,1}$ constraints. Sections 5 and 6 will demonstrate this point theoretically and empirically.

**Applications of Hawkes processes**: Somewhat related to our motivation, there are a few recent studies that pay particular attention to the duality between microscopic (instance-level) and macroscopic (type- or aggregate-level) random processes. For instance, Wang et al. [43] discuss the relationship between a point process and a rate equation at an aggregate level, Li et al. [19] discuss macroscopic information diffusion through human interactions, and Zhang et al. [49] discuss failure propagation in compressor stations. None of them, however, explicitly performs simultaneous instance- and type-level causal analysis. As discussed in Sec. 6, our framework can be viewed as a disciplined and effective solution to the issue of "warning fatigue," which prevails across many industries [10, 28, 7].

## 3  Preliminaries

This section provides the problem setting and recapitulates the basics of stochastic point processes.

### 3.1  Problem setting

Before getting into the formal definitions, let us take a brief look at a concrete use-case from cloud data center management (See Sec. 6 for more details). In a data center, various computer and network devices continuously produce numerous event logs, which can be viewed as marked temporal event data with "marks" being the warning types. Due to inter-connectivity of the devices, one event from a device, such as a warning of type "response time too long," may trigger many related events downstream. The more critical the original error, the more numerous the resulting set of events tend to be. Thus for a given event *instance* of interest (called the target), it is desirable to find which event instances in the past are causally related to it. We call this task *event causal diagnosis*.

There are two essential requirements in event causal diagnosis. The *first* one is the ability of providing instance-specific causal information in addition to the type-level causality. For instance, even if the $i$-th event type is *on average* likely to have a causal relationship with the $j$-th type, one specific *instance* of the $i$-th event type may have occurred spontaneously. A practical solution for event causal analysis must therefore perform type- and instance-level causal analysis simultaneously. The *second* requirement is the ability of providing *sparse* causal relationship. Since the number of event instances can be large, the capability of effectively shortlisting the candidates that may be causally related to a target event is essential in practice. To the best of our knowledge, our sparse Hawkes-Granger model is the first that meets these two requirements.

We are given an event sequence of $N + 1$ event *instances*:

$$\mathcal{D} = \{(t_0, d_0), (t_1, d_1), \ldots, (t_N, d_N)\}, \tag{1}$$

where $t_n$ and $d_n$ are the timestamp and the *event type* of the $n$-th event, respectively. The timestamps have been sorted in non-decreasing order $t_0 \le t_1 \le \cdots \le t_N$. There are $D$ event types $\{1, 2, \ldots, D\}$ with $D \ll N$. We take the first time stamp $t_0$ as the time origin. Hence, the remaining $N$ instances are thought of as realization of random variables, given $d_0$. As a general rule, we use $t$ or $u$ as a free variable representing time while those with a subscript denote an instance.

The main goal of event causal diagnosis, which is an *unsupervised learning task*, is to compute the instance *triggering probabilities* $\{q_{n,i}\}$, where $q_{n,i}$ is the probability for the $n$-th event ($n = 1, \ldots, N$)

instance to be triggered by the $i$-th event ($i = 0, \ldots, n$). By definition, $n \geq i$, and

$$\sum_{i=0}^{n} q_{n,i} = 1, \quad \forall n \in \{1, \ldots, N\}. \tag{2}$$

We call $q_{n,n}$ the *self* triggering (or simply self) probability. Note that providing $\{q_{n,i}\}$ amounts to providing weighted ranking of candidate triggering (or causing) events, where the weights sum to one. We wish to have as few candidates as possible with the aid of sparse causal learning.

## 3.2 Likelihood and intensity function

Since all the events are supposed to be correlated, the most general probabilistic model is the joint distribution of the $N$ events. By the chain rule of probability density functions (pdf), the joint distribution can be represented as

$$f((t_1, d_1), \ldots, (t_N, d_N) \mid (t_0, d_0)) = \prod_{n=1}^{N} f(t_n, d_n \mid \mathcal{H}_{n-1}),$$

where $\mathcal{H}_{n-1}$ denotes the event history up to $t_{n-1}$, namely, $\mathcal{H}_{n-1} \triangleq \{(t_0, d_0), \ldots, (t_{n-1}, d_{n-1})\}$. We use $f(\cdot)$ to symbolically denote a pdf. This decomposition readily leads to the definition of the base likelihood function $L_0$:

$$L_0 \triangleq \sum_{n=1}^{N} \ln f(t_n \mid d_n, \mathcal{H}_{n-1}) + \sum_{n=1}^{N} \ln f(d_n \mid \mathcal{H}_{n-1}). \tag{3}$$

The distribution $f(t \mid d_n, \mathcal{H}_{n-1})$ is defined on $t_{n-1} \leq t < \infty$ and satisfies the normalization condition in that domain. For the task of causal diagnosis, the first term in Eq. (3) plays the central role. We omit the second term in what follows, assuming $f(d_n \mid \mathcal{H}_{n-1})$ is a constant.

The intensity function given $\mathcal{H}_{n-1}$ is defined as the probability density that the first event since $t_{n-1}$ occurs. This is a conditional density. When considering the density at $t$, the condition reads "no event occurred in $[t_{n-1}, t)$." Hence,

$$\lambda_d(t \mid \mathcal{H}_{n-1}) \triangleq \frac{f(t \mid d, \mathcal{H}_{n-1})}{1 - \int_{t_{n-1}}^{t} \mathrm{d}u \, f(u \mid d, \mathcal{H}_{n-1})}, \tag{4}$$

where $\lambda_d(t \mid \mathcal{H}_{n-1})$ is the intensity function for the $d$-th event type, given the history $\mathcal{H}_{n-1}$. Notice that the r.h.s. can be written as $-\frac{\mathrm{d}}{\mathrm{d}t} \ln \left(1 - \int_{t_{n-1}}^{t} \mathrm{d}u \, f(u \mid d_n, \mathcal{H}_{n-1})\right)$. Integrating the both sides and arranging the terms, we have

$$f(t \mid d, \mathcal{H}_{n-1}) = \lambda_d(t \mid \mathcal{H}_{n-1}) \exp \left\{ -\int_{t_{n-1}}^{t} \mathrm{d}u \, \lambda_d(u \mid \mathcal{H}_{n-1}) \right\}, \tag{5}$$

which allows representing $L_0$ in terms of the intensity:

$$L_0 = \sum_{n=1}^{N} \left\{ \ln \lambda_{d_n}(t_n \mid \mathcal{H}_{n-1}) - \int_{t_{n-1}}^{t_n} \mathrm{d}u \, \lambda_{d_n}(u \mid \mathcal{H}_{n-1}) \right\}. \tag{6}$$

Note that the integral in the second term cannot be reduced to that of $(t_0, t_N)$ in general due to $d_n$ being dependent on $n$. This fact is sometimes ignored in the literature.

## 4 Cardinality-Regularized Hawkes-Granger Model

This section provides a specific model for the intensity function and discusses the connection to Granger causality to define the Hawkes-Granger model.

## 4.1 Intensity function and Granger causality

For the intensity function in Eq. (6), we introduce a specific parameterization of the Hawkes process:

$$\lambda_d(t \mid \mathcal{H}_{n-1}) = \mu_d + \sum_{i=0}^{n-1} A_{d,d_i} \phi_d(t - t_i). \tag{7}$$

where $\mu_d \geq 0$ and $\phi_d(t - t_i)$ are called the baseline intensity and the decay function, respectively, of the $d$-th type. $A_{d,d_i}$ is the $(d, d_i)$-element of a matrix $\mathsf{A} \in \mathbb{R}^{D \times D}$, which is called the impact matrix (a.k.a. triggering or kernel matrix).

The Hawkes process has potential indistinguishability issues in the second term due to the product form. We remove some of the indistinguishability by (i) imposing the normalization condition $\int_0^\infty \phi_d(u)\, \mathrm{d}u = 1$ and (ii) making $\phi_d$ independent of $d_i$. In this parameterizaton, $A_{d,d_i}$ is the triggering impact from event type $d_i$ to $d$ while $\phi_d$ represents the susceptibility of $d$. Also, $\mu_d$ represents the tendency of spontaneous occurrence. Due to the arbitrariness of the time unit, the decay function has the form $\phi_d(u) = \beta_d \varphi(\beta_d u)$, where $\varphi(\cdot)$ is a nondimensional function and $\beta_d$ is called the decay rate. Popular choices for $\varphi$ are the exponential $\exp(-u)$ and power $\eta(1 + u)^{-\eta-1}$ functions with $\eta > 1$ being a given constant.

Figure 2 illustrates the model (7), in which $\lambda_d(t \mid \mathcal{H}_4)$ is shown with an arbitrary decay function. We assume that $A_{d,d_1} = A_{d,d_3} = 0$ and $A_{d,d_2} > A_{d,d_4}$. The effect of the 2nd instance on $(t, d)$ is smaller than that of the 4th due to time decay despite the larger $A_{d,d_2}$. On the other hand, as shown with the dashed lines, the 1st and 3rd instances have no effect on the occurrence probability for the assumed $d$-th type in any future time point. This is in fact how Eichler et al. [9] defined the Granger non-causality in the Hawkes model (see also [1]):

**Definition 1** (Hawkes process and Granger non-causality [9])**.** *If $A_{d,d'} = 0$, event instances of the $d'$-type are Granger-non-causal to those of the $d$-th type.*

Definition 1 states that estimating $\mathsf{A}$ amounts to learning a Granger causal graph. This Granger non-cause characterization holds also in nonlinear Hawkes models, where the r.h.s. of Eq. (7) is replaced with $g(\mu_d + \sum_{i=0}^{n-1} A_{d,d_i} \phi_d(t - t_i))$, where $g(\cdot)$ is some nonlinear function. In that case, however, the original meaning of $\mu_d$ and $A_{d,d_i}$ no longer holds. For example, $\mu_d$ contributes not only to spontaneous occurrences but also to causal triggering, due to cross-terms be-

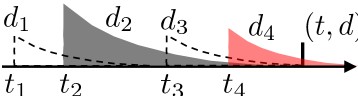

Figure 2: Illustration of Hawkes model in Eq. (7), showing $\lambda_d(t \mid \mathcal{H}_4)$ as an example. The $d_1$- and $d_3$-types are not causally related to $d$.

tween $\mu_d$ and $A_{d,d_i} \phi_d$ in the intensity function. Also, as the MM strategy is no longer applicable, there is no natural way of defining the triggering probabilities (see the next subsection). Given that our main interest lies in causal diagnosis ("who caused this?"), rather than black-box data fitting, the linear additive form Eq. (7) is particularly convenient, and hence is our primary model.

## 4.2 Cardinality-regularized minorization-maximization framework

As we saw in Fig. 2, achieving sparsity in $\mathsf{A}$ is of critical importance in instance-wise causal analysis: It directly leads to reducing the number of event candidates to be causally associated. To guarantee sparsity, we propose the following *cardinality-regularized* maximum likelihood:

$$\max_{\mathsf{A},\boldsymbol{\mu},\boldsymbol{\beta}} \left\{ L_0(\mathsf{A},\boldsymbol{\mu},\boldsymbol{\beta}) - \tau\|\mathsf{A}\|_0 - R_2(\mathsf{A},\boldsymbol{\mu},\boldsymbol{\beta}) \right\}, \quad R_2 \triangleq \frac{1}{2}\left(\nu_\mu\|\boldsymbol{\mu}\|_2^2 + \nu_\beta\|\boldsymbol{\beta}\|_2^2 + \nu_A\|\mathsf{A}\|_{\mathrm{F}}^2\right) \tag{8}$$

where we have defined $\boldsymbol{\beta} \triangleq (\beta_1, \ldots, \beta_D)^\top$ and $\boldsymbol{\mu} \triangleq (\mu_1, \ldots, \mu_D)^\top$ and the $\ell_0$ norm $\|\mathsf{A}\|_0$ represents the cardinality (the number of nonzero elements) of $\mathsf{A}$. Also, $\|\cdot\|_2$ is the 2-norm and $\|\cdot\|_{\mathrm{F}}$ is the Frobenius norm. $\tau, \nu_\beta, \nu_\mu, \nu_A$ are constants for regularization strength. For $\tau$, we note that Eq. (15) can be viewed as MAP (maximum a posteriori) estimation with the Bernoulli prior $(1-\gamma)^{\|\mathsf{A}\|_0} \gamma^{D^2 - \|\mathsf{A}\|_0}$, where $0.5 < \gamma < 1$ is the probability of getting 0 in the matrix elements. By taking the logarithm and equating to $-\tau\|\mathsf{A}\|_0$, we have

$$\tau = \ln[\gamma/(1-\gamma)]. \tag{9}$$

With $\gamma$ having the specific interpretation, this equation makes the choice of a $\tau$ value easier.

As mentioned earlier, numerically solving for MLE is known to be challenging even when $\tau = 0$, mainly due to the log-sum term $\ln \lambda_d$ in Eq. (6). The MM algorithm leverages the additive structure of the Hawkes process in Eq. (7) to apply Jensen's inequality in a manner similar to the EM algorithm for mixture models [30]. Specifically, we rewrite Eq. (7) as $\lambda_{d_n}(t_n|\mathcal{H}_{n-1}) = \sum_{i=0}^{n} \Phi_{n,i}^{d_n,d_i}$, where

$$\Phi_{n,i}^{d_n,d_i} \triangleq \begin{cases} \mu_{d_n}, & i = n \\ A_{d_n,d_i}\phi_d(t_n - t_i), & i = 0, \ldots, n-1. \end{cases} \tag{10}$$

With an arbitrary distribution $q_{n,i}$ over $i$ such that $\sum_{i=0}^{n} q_{n,i} = 1$ for $\forall n$, Jensen's inequality guarantees $\ln \sum_{i=0}^{n} \Phi_{n,i}^{d_n,d_i} \geq \sum_{i=0}^{n} q_{n,i} \ln \frac{\Phi_{n,i}^{d_n,d_i}}{q_{n,i}}$, which leads to a lower bound of the log likelihood:

$$L_0 \geq L_1 \triangleq \sum_{n=1}^{N} \left\{ \sum_{i=0}^{n} q_{n,i} \ln \frac{\Phi_{n,i}^{d_n,d_i}}{q_{n,i}} - \mu_{d_n}\Delta_{n,n-1} - \sum_{i=0}^{n-1} A_{d_n,d_i} \int_{\Delta_{n-1,i}}^{\Delta_{n,i}} du\ \phi_{d_n}(u) \right\}, \tag{11}$$

where we defined $\Delta_{n,i} \triangleq t_n - t_i$. The tightest bound is obtained by maximizing the r.h.s. with respect to $q_{n,i}$ under the normalization condition:

$$q_{n,i} = \begin{cases} \lambda_{d_n}(t_n|\mathcal{H}_{n-1})^{-1}\mu_{d_n}, & i = n, \\ \lambda_{d_n}(t_n|\mathcal{H}_{n-1})^{-1}A_{d_n,d_i}\phi_{d_n}(t_n - t_i), & i \neq n. \end{cases} \tag{12}$$

With a proper initialization, the MM algorithm estimates $\{q_{n,i}\}$ and $(\boldsymbol{\mu}, \boldsymbol{\beta}, \mathsf{A})$ alternately. The whole procedure is concisely summarized as

$$\boldsymbol{\mu}, \boldsymbol{\beta}, \mathsf{A} = \arg\max\left\{L_1 - \tau\|\mathsf{A}\|_0 - R_2\right\}, \quad \text{given} \quad \{q_{n,i}\}. \tag{13}$$

$$\{q_{n,i}\} = (\text{Eq. (12)}), \quad \text{given} \quad \boldsymbol{\mu}, \boldsymbol{\beta}, \mathsf{A}. \tag{14}$$

Similarity to the EM algorithm [30] is evident, but they differ since we apply Jensen's inequality only partially in $L_0$. In this framework, $\{q_{n,i}\}$ was introduced as a mathematical artifact in Jensen's inequality. However, it opens a new door to instance-level causal analysis. We interpret $q_{n,i}$ as the *instance triggering probability* that the $n$-th instance has been triggered by the $i$-th instance. The MM-based Hawkes process is causal both at the instance-level and the type-level. With this in mind, we call the framework of Eqs. (13)-(14) the *Hawkes-Granger model*.

## 5    Sparse Causal Learning via Cardinality Regularization

This section discusses how to find the solution $\mathsf{A}$ in Eq. (13). We leave the estimation procedure of $\boldsymbol{\mu}$ and $\boldsymbol{\beta}$ to Appendix A (all the Appendices are in the supplementary material).

By Eqs. (8) and (11), the optimization problem of Eq. (13) with respect to $\mathsf{A}$ is rewritten as

$$\max_{\mathsf{A}} \left\{ \sum_{k,l=1}^{D} (Q_{k,l} \ln A_{k,l} - H_{k,l}A_{k,l}) - \frac{\nu_A}{2}\|\mathsf{A}\|_{\mathrm{F}}^2 - \tau\|\mathsf{A}\|_0, \right\}, \tag{15}$$

where we have defined matrices $\mathsf{Q}$ and $\mathsf{H}$ as

$$Q_{k,l} \triangleq \sum_{(n,i)} \delta_{d_n,k}\delta_{d_i,l}q_{n,i}, \quad H_{k,l} \triangleq \sum_{(n,i)} \delta_{d_n,k}\delta_{d_i,l}h_{n,i}, \quad h_{n,i} \triangleq \int_{\Delta_{n-1,i}}^{\Delta_{n,i}} du\ \phi_{d_n}(u). \tag{16}$$

$Q_{k,l}$ represents how likely the type $k$ and $l$ become a cause-effect pair. For ease of exposition, consider the vectorized version of the problem by defining $\boldsymbol{x} \triangleq \mathrm{vec}\,\mathsf{A}$, $\boldsymbol{h} \triangleq \mathrm{vec}\,\mathsf{H}$, and $\boldsymbol{g} \triangleq \mathrm{vec}\,\mathsf{Q}$:

$$\max_{\boldsymbol{x}} \left\{ \sum_m \Psi_m(x_m) - \tau\|\boldsymbol{x}\|_0 \right\}, \quad \Psi_m(x_m) \triangleq \left( g_m \ln x_m - h_m x_m - \frac{\nu_A}{2}x_m^2 \right), \tag{17}$$

where $g_m, h_m \geq 0$, $\tau, \nu_A > 0$ hold. This is the main problem we consider in this section.

Let us first look at what would happen if we instead used the popular $\ell_1$ or $\ell_{2,1}$ regularizer here. For the $\ell_p$ norm $\|\boldsymbol{x}\|_p \triangleq (\sum_m |x_m|^p)^{\frac{1}{p}}$, the following theorem holds:

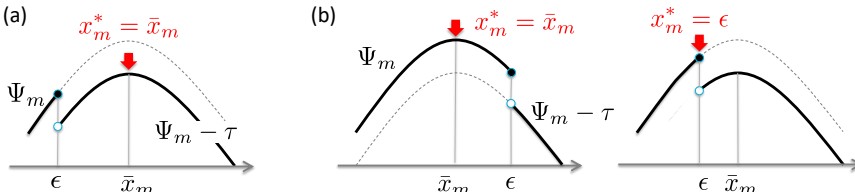

Figure 3: Three cases in Eq. (18): (a) $x_m^* > \epsilon$ and (b) two possibilities when $x_m^* \leq \epsilon$.

**Theorem 1.** *For $p \geq 1$, the problem $\max_{\boldsymbol{x}} \left\{ \sum_m \Psi_m(x_m) - \tau \|\boldsymbol{x}\|_p \right\}$ is convex and has a unique solution. Let $\boldsymbol{x}^{**}$ be the solution. The solution cannot be sparse, i.e., $x_m^{**} \neq 0$ for $\forall m$, if $g_m > 0$.*

(*Proof*) The convexity follows from $\frac{\mathrm{d}^2}{\mathrm{d}x_m^2} \Psi(x_m) = -(g_m/x_m^2) - \nu_A < 0$ and the convexity of the $\ell_p$-norm. The second statement follows from $\lim_{x \to 0} \ln x = -\infty$. $\square$

In the MM iteration, we need to start with some $Q_{k,l} > 0$ for all $k, l$ to make all the event types eligible in causal analysis. In that case, $\forall m$, $g_m > 0$, and thus any $A_{k,l}$ cannot be zero. Therefore, under any $p \geq 1$, including $p = 1$ ($\ell_1$) and $2$ ($\ell_{2,1}$), any "sparse" solution must be viewed as a numerical artifact. One can avoid this particular issue by performing conventional MLE without the MM algorithm, but, as discussed earlier, that is not a viable solution in practice due to the well-known numerical difficulties of the conventional MLE [42, 26].

This situation is reminiscent of the issue with density estimation with probabilistic mixture models as discussed by Phan and Idé [35], who first introduced the notion of "$\epsilon$-sparsity." Here we employ a similar approach. To handle the singularity at zero, we introduce a small constant $\epsilon > 0$ and we propose to solve

$$\boldsymbol{x}^* = \max_{\boldsymbol{x}} \sum_m \left\{ \Psi_m(x_m) - \tau I(x_m > \epsilon) \right\}, \tag{18}$$

to get an $\epsilon$-sparse solution instead of the original problem (17), where $I(\cdot)$ is the indicator function that returns 1 when the argument is true and 0 otherwise. Note that the condition $\nu_A > 0$ is important for stable learning when some $g_m$ are close to zero; It makes the objective strongly convex (easily seen from the proof of Theorem 1) and thus makes the problem (17) well-behaved. We remark that while it is possible to use an analogous $\epsilon$-sparsity approach with the $\ell_1$ or $\ell_{2,1}$ regularizer, it may not be as well-behaved due to the piece-wise linearity of $\ell_1$-norm.

To solve Eq. (18), we first note that $\Psi_m$ is concave and has the maximum at

$$\bar{x}_m \triangleq \frac{1}{2\nu_A} \left( -h_m + \sqrt{h_m^2 + 4\nu_A g_m} \right), \tag{19}$$

which is obtained by solving $\frac{\mathrm{d}\Psi_m}{\mathrm{d}x_m} = 0$. Suppose that we had a solution of Eq. (18) somehow, and let us define a set of indices $\mathcal{B} \triangleq \{m \mid x_m^* > \epsilon\}$. Since the objective function in Eq. (18) gets a penalty $-\tau$ in $x_m > \epsilon$, for an $x_m$ to be able to be the solution in this domain, the objective must catch up on $\tau$ at its maximum (see Fig. 3 (a) for illustration). Hence, $\forall m \in \mathcal{B}$, we have

$$\bar{x}_m > \epsilon, \quad \Psi_m(\epsilon) < \Psi_m(\bar{x}_m) - \tau. \tag{20}$$

Conversely, if these conditions are met, $m$ must be in $\mathcal{B}$ due to the concavity of $\Psi_m$. Therefore, we can learn whether an index $m$ is in $\mathcal{B}$ or not by checking these conditions.

With this fact in mind, we solve Eq. (18) for $m \in \mathcal{B}$ and $\notin \mathcal{B}$ separately:

$$\forall m \in \mathcal{B}, \quad x_m^* = \arg\max_{x_m} \left\{ \Psi_m(x_m) - \tau \right\}, \tag{21}$$

$$\forall m \notin \mathcal{B}, \quad x_m^* = \arg\max_{x_m} \Psi_m(x_m) \quad \text{subject to} \quad \epsilon - x_m \geq 0. \tag{22}$$

For $\forall m \in \mathcal{B}$, the optimality condition is

$$\frac{\mathrm{d}}{\mathrm{d}x_m} \left\{ \Psi_m(x_m) - \tau \right\} = \frac{g_m}{x_m} - h_m - \nu_A x_m = 0, \tag{23}$$

**Algorithm 1** $L_0$`Hawkes`: sparse learning impact matrix A
___
1: **Input:** $\boldsymbol{g} = \text{vec } \mathsf{Q}$, $\boldsymbol{h} = \text{vec } \mathsf{H}$, and $\nu_A > 0, \tau \geq 0, \epsilon > 0$.
2: **for all** $m = 1, \ldots, D^2$ **do**
3:     compute $\bar{x}_m$ with Eq. (19).
4:     check Eq. (20) to see if $m \in \mathcal{B}$.
5:     **if** $m \in \mathcal{B}$ **then**
6:         $x_m^* = \bar{x}_m$
7:     **else**
8:         $x_m^* = \min\{\epsilon, \bar{x}_m\}$
9:     **end if**
10: **end for**
11: **return** $\mathsf{A}^* = \text{vec}^{-1} \boldsymbol{x}^*$ (i.e., convert back to matrix)
___

which readily gives $x_m^* = \bar{x}_m$ under Eq. (20).

For $\forall m \notin \mathcal{B}$, with a Lagrange multiplier $\xi_m$, the Karush–Kuhn–Tucker (KKT) condition is given by

$$\frac{g_m}{x_m} - h_m - \nu_A x_m - \xi_m = 0, \quad \xi_m(\epsilon - x_m) = 0, \quad \epsilon \geq x_m, \quad \xi_m \geq 0. \tag{24}$$

As illustrated in Fig. 3 (b), there are two possibilities here: One is $x_m^* \neq \epsilon$. In this case, $\xi_m$ must be 0, and the first equation gives $x_m^* = \bar{x}_m$, which holds when $\bar{x}_m \leq \epsilon$. The other is $x_m^* = \epsilon$, which holds when $\bar{x}_m > \epsilon$.

Although, $\epsilon$ can be viewed as a 'zero-ness' threshold, the solution derived is qualitatively different from naive thresholding applied to the non-regularized solution $\bar{x}$. See Sec. 6 for an empirical validation.

**Algorithm summary**    Algorithm 1 summarizes $L_0$`Hawkes`, the proposed algorithm, which is used as part of the iterative MM procedure in Eq. (13). The estimation procedure for $\boldsymbol{\mu}, \boldsymbol{\beta}$ can be found in Appendix A. The total complexity is $\mathcal{O}(N^2 + D^2)$, which is the same as for the existing MM algorithm. For input parameters, $\tau$ can be fixed to a value of user's choice in $0.5 < \gamma < 1$ through Eq. (9). The parameters $\nu_A, \nu_\beta, \nu_\mu$ are important for stable convergence. It is recommended to start with a small positive value, such as $10^{-5}$, and increase it if numerical issues occur. These parameters should eventually be cross-validated with independent episodes of event data, or ground through causality data. In Sec. 6, we presents an approach to determine $\epsilon$ as the one that gives the break-even accuracy. If validation dataset is unavailable at all, the use of Akaike's information criterion (AIC) can be one viable approach, given that $\|\mathsf{A}\|_0$ approximates the total number of free parameters fitted.

## 6    Experiments

Our focus in this section is to (1) show the impact of the equi-time-interval assumption of the existing Granger causal learning models, (2) demonstrate how $L_0$`Hawkes` produce a sparse solution, and (3) show its utility in real-world use-cases. We leave the detail of the experimental setup to Appendix B.

**Comparison with neural Granger models**. We generated two synthetic multivariate event datasets, `Sparse5` and `Dense10`, with a standard point process simulator `tick` [4]. `Sparse5` has $D = 5$ with a sparse causal graph shown in Fig. 1 (b). We generated 5 different datasets by changing the random seed of `tick`, one of which has been shown in Fig. 1 (c). `Dense10` has $D = 10$ and was generated with a relatively dense and noisy causal graph. Both have $N \sim 1\,000$ event instances.

First, we compared $L_0$`Hawkes` with `cLSTM` and `cMLP` [40], state-of-the-art Granger causal learning algorithms, on `Sparse5`. These correspond to a nonlinear extension of autoregressive and state-space models, respectively, covering the two main time-series prediction paradigms known to date. We estimated A or the Granger causal matrix with many different values of the regularization parameters. We chose $\nu_A, \nu_\beta, \nu_\mu$ to be 0.1 and tested $\tau = 0.5, 1, 2$. The goal is to retrieve the simple causal graph in Fig. 1 (b), which has 2 positive and 18 negative edges as the ground truth, omitting the self-loops. For `cLSTM` and `cMLP`, the dataset was converted into 5-dimensional equi-interval time-series of event counts with $N$ time points, based on a sliding window whose size is $w = 10$ times larger than the mean event inter-arrival time. The number of hidden units and the lag (or the context length of LSTM)

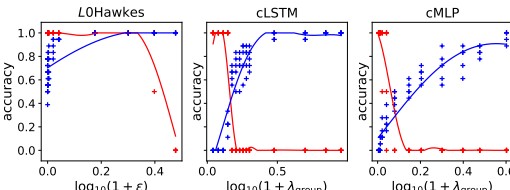

Figure 4: TN (red) and TP (blue) accuracies as a function of log regularization strength.



Figure 5: Comparison of $x^*$ (flattened A in each row) computed with 100 different $\tau$ values.

were set to 100 (from [40]) and 10, respectively, for both. The mean computational time was (46, 881, 382) seconds per one parameter set for ($L_0$Hawkes, cMLP, cLSTM), respectively, on a laptop PC (i7 CPU, 32GB memory, Quadro P3200 GPU).

Figure 4 compares the true positive (TP) and true negative (TN) accuracies as a function of log regularization strength. $L_0$Hawkes uses $\tau = 1$. The plot, which we call the *contrastive accuracy*

Table 1: Break-even accuracies in Fig. 4.

| $L_0$Hawkes | cLSTM | cMLP |
|---|---|---|
| $1.00 \pm 0.11$ | $0.43 \pm 0.09$ | $0.31 \pm 0.10$ |

*plot* (CAP), allows us to directly choose one regularization strength and is more useful than the ROC (receiver operating characteristic) curve. This is especially true in this case, where the variety of positive samples is limited. See Appendix B for more comments on CAP. The fitting curves in the CAP were computed with Gaussian process regression (GPR) [37] with optimized hyperparameters. The TP accuracy at the intersection with the TN curve are called the *break-even accuracy*, which can be thought of as an overall performance metric. Table 1 summarizes the break-even accuracies, where the standard deviation was estimated with the GPR fitting. The contrast is clear. cLSTM and cMLP failed to capture the very simple causality while $L_0$Hawkes reproduced it almost perfectly. The main reason is that the event density of self-exciting events can be highly non-uniform in time (as seen from Fig. 1 (c)), which is inconsistent to the equi-interval assumption of the autoregressive or RNN-type models. The conclusion held over all the parameter values we tested: $\tau \in \{0.5, 1, 2\}$ and $w \in \{1, 2, 5, 10, 20\}$.

**Comparison with sparse Hawkes models**. Next, we compared $L_0$Hawkes with $\ell_1$-regularizatoin based [51] ('L$_1$') and $\ell_{2,1}$-regularization based [46] ('L$_{2,1}$') models on Dense10. Figure 5 compares the solution A$^*$ obtained by the three methods. We solved Eq. (17) 100 times for each, changing the value of the regularization strength $\tau$ from 0 to 2. We used $\epsilon = 0.01$ for $L_0$Hawkes. All the three methods share the same values of $\tau$ and $\nu_A = 10^{-9}$. The matrix elements are sorted in the ascending order of the value of $\bar{x}$ in Eq. (19). As shown in the figure, L$_1$ and L$_{2,1}$ produce a smooth *non*-sparse profile, as expected from Theorem 1. In contrast, $L_0$Hawkes gets more "off" entries (shown in yellow) as $\tau$ increases. As clearly seen from the irregular sparsity patterns in Fig. 5, the result is *different from naive thresholding on $\bar{x}$* (Eq. (19)). At a given sparsity level, our solution is guaranteed to achieve the highest possible likelihood while naive thresholding is not.

Now that we have confirmed the capability of $L_0$Hawkes in sparse Granger causal learning, let us leverage it in real-world causal diagnosis tasks.

**Power grid failure diagnosis**. We obtained failure event data ('Grid') of power grid from U.S. Department of Energy [29]. The failure events represent abrupt changes in the voltage and/or current signals measured with phasor measurement units (PMUs), which are deployed in geographically distributed locations in the power grid. The network topology is not given for privacy concerns. Only anonymized PMU IDs are given. We are interested in discovering a hidden causal relationship in a data-driven manner from the temporal event data alone. This is a type-level causal diagnosis task.

The dataset records $N = 3\,811$ failure events labeled as "line outages" from $D = 22$ PMUs over a 10-month period. We grid-searched the model parameters based on AIC to get $5 \times (10^{-3}, 10^{-4}, 10^{-4})$ for $(\nu_\mu, \nu_\beta, \mu_A)$ and $(1, 1)$ for $(\tau, \epsilon)$. The value of $\epsilon$ corresponds to about 3% of $\max_{k,l} A_{k,l}$. We used the same $\tau$ for the $\ell_1$ and $\ell_{2,1}$ regularizers. We used the power decay of $\eta = 2$ to capture long-tail behaviors. Figure 6 compares computed A, in which nonzero matrix elements are shown in black. In L$_1$ and L$_{2,1}$, zero entries can appear only when $Q_{k,l}$ happens to be numerically zero. In contrast, $L_0$Hawkes enjoys guaranteed sparsity. From the computed A, a hidden causal structure among PMUs were successfully discovered. In particular, a PMU called B904 seems to be a dominatingly influential source of failures. We leave further details to another paper.

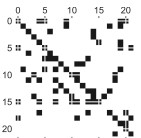 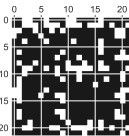 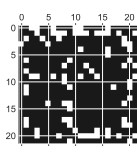 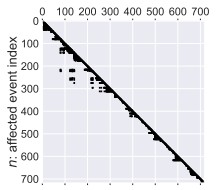 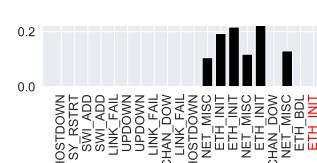

Figure 6: Sparsity pattern of estimated A on the Grid data with $L_0$Hawkes (left), L$_1$ (middle), and L$_{2,1}$ (right).

Figure 7: Results on the Cloud data. (Left) Nonzero elements of $\{q_{n,i}\}$. (Right) Triggering probabilities for the 150th instance (i.e. $q_{150,i}$).

**Data center warning event diagnosis**. Finally, we applied $L_0$Hawkes to a real data center management task. We obtained $N = 718$ warning events from a real cloud system. These events result from filtering logs emitted by network devices and each has its type. There are $D = 14$ unique event types in our dataset ('Cloud'). This is an *instance-level* causal diagnosis task ("who caused this?").

Figure 7 (left) visualizes nonzero entries of $\{q_{n,i}\}$, where those with $q_{n,i} < 0.01$ are omitted. As expected, $\{q_{n,i}\}$ is quite sparse, and hence, event consolidation can be straightforwardly performed by picking nonzero triggering probabilities. The right panel shows such an example, showing $q_{150,i}$, in which the rightmost slot (in red) corresponds to the self probability $q_{150,150}$. For each $i$, its event type $d_i$ is shown below the bar. The type of the event in question, ETH_INIT, is related to the process of initializing an Ethernet interface. Note in the figure that the self probability of this instance was computed as 0, while several preceding instances of the same type have positive triggering probabilities, leading to successful suppression of duplication.

Many instances have zero triggering probability despite their time proximity (the six events with positive probabilities were within 27 seconds from the 150th event), thanks to the sparsity of A. For example, this dataset contains 416 instances of event type UPDOWN adding considerable noise but were appropriately ignored by our method. Unlike naive hard-windowing approaches, our framework is able to sift for genuine causal relationships.

## 7  Concluding remarks

We proposed a new sparse Granger-causal learning framework for stochastic event data. We pointed out that the existing sparse MM algorithms do not have a sparse solution in the exact sense (Theorem 1). We also showed that the existing neural Granger approaches (cMLP, cLSTM) are of limited use for event data, mainly due to its equi-time-interval assumption. The proposed Hawkes-Granger model produces mathematically well-defined sparsity and allows simultaneous instance- and type-level causal event diagnosis with good interpretability enabled by the sparsity.

**Acknowledgement**   T.I. is partially supported by the Department of Energy National Energy Technology Laboratory under Award Number DE-OE0000911. A part of this report was prepared as an account of work sponsored by an agency of the United States Government. Neither the United States Government nor any agency thereof, nor any of their employees, makes any warranty, express or implied, or assumes any legal liability or responsibility for the accuracy, completeness, or usefulness of any information, apparatus, product, or process disclosed, or represents that its use would not infringe privately owned rights. Reference herein to any specific commercial product, process, or service by trade name, trademark, manufacturer, or otherwise does not necessarily constitute or imply its endorsement, recommendation, or favoring by the United States Government or any agency thereof. The views and opinions of authors expressed herein do not necessarily state or reflect those of the United States Government or any agency thereof.

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
