# Cardinality-Regularized Hawkes-Granger Model

**Tsuyoshi Idé**
IBM Research, T. J. Watson Research Center
tide@us.ibm.com

**Georgios Kollias**
IBM Research, T. J. Watson Research Center
gkollias@us.ibm.com

**Dzung T. Phan**
IBM Research, T. J. Watson Research Center
phandu@us.ibm.com

**Naoki Abe**
IBM Research, T. J. Watson Research Center
nabe@us.ibm.com

## Appendix / supplementary material

## A   Solutions for Baseline Intensity and Decay Parameters

This section provides parameter estimation equations in the MM procedure Eq. (13) for the baseline intensity $\boldsymbol{\mu}$ and the decay parameter $\boldsymbol{\beta}$, which were omitted in the main text due to space limitations. Our goal is to find the maximizer of the objective function

$$L(\boldsymbol{\mu}, \boldsymbol{\beta}) \triangleq L_1 - \frac{1}{2} \left( \nu_\mu \|\boldsymbol{\mu}\|_2^2 + \nu_\beta \|\boldsymbol{\beta}\|_2^2 \right), \tag{A.1}$$

where $L_1$ has been defined by Eq. (11) in the main text.

For the baseline intensity, by collecting all the related terms in $L$, the objective function to be maximized is given by

$$L = \sum_{n=1}^{N} \left\{ q_{n,n} \ln \frac{\mu_{d_n}}{q_{n,n}} - \mu_{d_n} \Delta_{n,n-1} \right\} - \frac{1}{2} \nu_\mu \|\boldsymbol{\mu}\|_2^2 + \text{const.}, \tag{A.2}$$

where $\Delta_{n,n-1} \triangleq t_n - t_{n-1}$. By differentiating w.r.t. $\mu_k$, the maximizer can be straightforwardly obtained as

$$\mu_k = \frac{1}{2\nu_\mu} \left( -D_k^\mu + \sqrt{(D_k^\mu)^2 + 4\nu_\mu N_k^\mu} \right), \tag{A.3}$$

where

$$D_k^\mu = \sum_{n=1}^{N} \delta_{d_n,k} \Delta_{n,n-1}, \quad N_k^\mu = \sum_{n=1}^{N} \delta_{d_n,k} q_{n,n} \tag{A.4}$$

with $\delta_{d_n,k}$ being Kronecker's delta.

For the decay parameter, the objective function becomes

$$L = \sum_{n=1}^{N} \sum_{i=0}^{n-1} \left\{ q_{n,i} \ln \frac{\phi_{d_n}(\Delta_{n,i})}{q_{n,i}} - A_{d_n,d_i} h_{n,i} \right\} - \frac{1}{2} \nu_\beta \|\boldsymbol{\beta}\|_2^2 + \text{const}, \tag{A.5}$$

where $h_{n,i} \triangleq \int_{\Delta_{n-1,i}}^{\Delta_{n,i}} \mathrm{d}u \, \phi_{d_n}(u)$. In this case, the maximizer depends of a specific choice of the decay function. The general form of the solution is given by

$$\beta_k = \frac{1}{2\nu_\beta} \left( -D_k^\beta + \sqrt{(D_k^\beta)^2 + 4\nu_\beta N_k^\beta} \right), \quad N_k^\beta = \sum_{n=1}^{N} \delta_{d_n,k}(1 - q_{n,n}). \tag{A.6}$$

35th Conference on Neural Information Processing Systems (NeurIPS 2021).

Below, we provide results for the exponential and power distributions. For the exponential distribution, we have

$$D_k^\beta = \sum_{n=1}^{N} \delta_{d_n,k} \sum_{i=0}^{n-1} \left[ q_{n,i} \Delta_{n,i} + A_{k,d_i} \frac{\partial h_{n,i}}{\partial \beta_k} \right], \qquad (A.7)$$

$$\frac{\partial h_{n,i}}{\partial \beta_k} = \delta_{d_n,k} \left[ \Delta_{n,i} \mathrm{e}^{-\beta_k \Delta_{n,i}} - \Delta_{n-1,i} \mathrm{e}^{-\beta_k \Delta_{n-1,i}} \right]. \qquad (A.8)$$

For the power distribution, we have

$$D_k^\beta = \sum_{n=1}^{N} \delta_{d_n,k} \sum_{i=0}^{n-1} \left[ \frac{(\eta+1) q_{n,i} \Delta_{n,i}}{1 + \beta_k \Delta_{n,i}} + A_{k,d_i} \frac{\partial h_{n,i}}{\partial \beta_k} \right], \qquad (A.9)$$

$$\frac{\partial h_{n,i}}{\partial \beta_k} = \delta_{k,d_n} \left\{ \frac{\eta \Delta_{n,i}}{(1 + \beta_k \Delta_{n,i})^{\eta+1}} - \frac{\eta \Delta_{n-1,i}}{(1 + \beta_k \Delta_{n-1,i})^{\eta+1}} \right\}. \qquad (A.10)$$

# B  Experimental Details

This section describes the details of the experiments. We have included the `Sparse5` and `Dense10` data sets and the Python code to generate those as part of the final submission.

## B.1  Data generation

`Sparse5`  The `Sparse5` benchmark dataset is designed to have a simplest but nontrivial kind of causal structure, which is supposed to be easily reproduced by any Granger-causal learning algorithms. Using a standard point process simulator `tick` [1][1], we generated `Sparse5` by giving 0.001 to `baseline` for all the types and

$$\texttt{decays} = \begin{pmatrix} 0.5 \\ 0.5 \\ 0.1 \\ 0.1 \\ 0.1 \end{pmatrix} \mathbf{1}_5^\top, \quad \texttt{adjacency} = \begin{pmatrix} 0. & 0. & 0. & 0. & 0. \\ 1.5 & 0. & 0. & 0. & 0. \\ 0. & 0. & 0. & 0. & 0. \\ 0. & 0. & 1.5 & 0. & 0. \\ 0. & 0. & 0. & 0. & 0.75 \end{pmatrix}, \qquad (B.11)$$

where $\mathbf{1}_5$ is the 5-dimensional vector of ones and we employed the exponential distribution as the decay function. The numbers above were manually adjusted so `tick` did not produce a "spectral radius error" and all the event types have roughly the same number of event instances. In `tick`, we provided `seed` $= 1, 2, 3, 4, 5$, which resulted in five realizations of the 5-dimensional point process as shown in Fig. B.1. Due to the stochastic nature, the total number of event instances cannot be controlled. We manually adjusted the duration of simulation so the total number of events is roughly $N \approx 1\,000$. In particular, we had $N = (966, 991, 978, 960, 1030)$ for `seed` $= (1, 2, 3, 4, 5)$, respectively.

`Dense10`  The second benchmark dataset `Dense10` was generated again with `tick` [1]. We generated $N = 1\,121$ random events with $D = 10$. For parameters, we set `decay` $= 10$ (exponential decay) and `baseline` $= 1$ for all event types. For the impact matrix, we first randomly generated a binary matrix so that about half of the entries get 1 and at least one node is dense. Then, to simulate real-world noise, we added gamma-distributed noise with the shape and scale both being 1, as shown in Fig. B.2 (top left). See the attached code for the detail. For each generated event instance pair, we computed $\{q_{n,i}\}$ and $\{h_{n,i}\}$ to eventually get $\mathsf{Q}$ and $\mathsf{H}$. Figure B.2 shows what $\mathsf{Q}, \mathsf{H}$ look like. Figure B.2 also shows $[Q_{k,l}/H_{k,l}]$, which corresponds to non-sparse solution ($\tau \to 0$) in $\nu_A \to 0_+$. As shown in the figure, except for the scale, the overall pattern of $[Q_{k,l}/H_{k,l}]$ is quite similar to that of A.

## B.2  Performance metric and contrastive accuracy plot

In the main text, we drew the true positive (TP) and true negative (TN) accuracies as a function of the logarithmic regularization strength. There are multiple definitions of TP and TN accuracies in the

---

[1] `https://x-datainitiative.github.io/tick/`

Figure B.1: `Sparse5` dataset corresponding to $\mathtt{seed} = 1, 2, 3, 4, 5$, from top to bottom in the `tick` simulator.

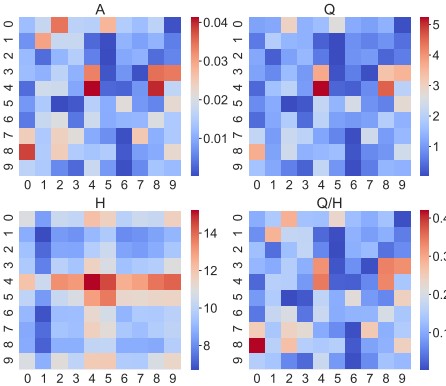

Figure B.2: `Dense10`: Randomly generated A with added gamma noise (top left), which was used to generate temporal events. From the event data, Q (top right), H (bottom left), and $[Q_{k,l}/H_{k,l}]$ (bottom right) were computed.

literature. Our definition is as follows:

$$\text{(TP accuracy)} = \frac{\text{(The number of successfully predicted actually positive samples)}}{\text{(The number of actually positive samples)}} \qquad (B.12)$$

$$\text{(TN accuracy)} = \frac{\text{(The number of successfully predicted actually negative samples)}}{\text{(The number of actually negative samples)}} \qquad (B.13)$$

In our setting, we thought of the nonzero off-diagonal elements in the ground truth Granger causal matrix (Fig. 1 (b)) as the actually positive samples. Similarly, the zero off-diagonal elements in the ground truth graph are defined as the actually negative samples.

The TP and TN accuracy values depend on the decision threshold to discriminate between positive and negative. In our setting, regularization strengths play the role of the threshold. We used a plot showing both the TP and TN accuracies as a function of the threshold parameter to evaluate the overall performance. Such a plot can be called the *contrastive accuracy plot* (CAP). We believe that CAP is more useful than the ROC curve, especially when a significant disproportion exists between the numbers of samples in positive and negative classes. Specifically, first, unlike ROC, CAP provides a direct way of choosing the threshold value yielding the best performance. Second, CAP has the flexibility of applying a transformation to the threshold value for better visualization. In our case, we

used a log-transformed regularization strength to make the curves look smoother. Third, unlike the AUC (area-under-the-curve), which typically requires numerical integration to get the number, CAP immediately provides the break-even accuracy, the intersection between the TP and TN curves, as an easy-to-consume overall performance metric. Finally, we note that CAP has the same information as ROC because they both draw a trajectory of the accuracies over the entire set of the threshold values.

To handle the variability due to the stochastic fluctuations over the five datasets in `Sparse5`, we used the Gaussian process regression. Here, the input variable is a regularization strength (symbolically denoted by $\lambda$) and the target variable is the TP or TN accuracy. We used the Gaussian kernel (a.k.a. RBF (radial basis function)) with a constant shift defined as

$$k(\lambda_i, \lambda_j) = C_0 + C \exp \left\{ -\frac{1}{2L^2}(\lambda_i - \lambda_j)^2 \right\}, \tag{B.14}$$

where $\lambda_i, \lambda_j$ are instances of $\lambda$. The hyper-parameters $C_0, C, L$ are optimized by maximizing the log marginalized likelihood. Predicted values greater than 1 (smaller than 0) were replaced with 1 (0), respectively. We used the `GaussianProcessRegressor` module[2] of `scikit-learn`.

### B.3 Running neural Granger methods

The neural Granger learning methods `cLSTM` and `cMLP` [3] take real-valued time-series data as the input. A widely-used approach to converting a marked event sequence to a multivariate time-series in practice is window-based counting (see, e.g. [2]). Specifically, the number of occurrences within the sliding window is recorded as a time-series value for each event type. This is a reasonable approach from the perspective of the point process theory because many event sequences can be modeled with an inhomogeneous Poisson process to a reasonable approximation and Poisson processes as the *counting process* are described with the count as the sufficient statistic.

To determine the window size, we first calculated the mean inter-event arrival time $\langle dt \rangle$ from the original event data, and set up a sliding window of size $w = 10\langle dt \rangle$ in the main text as well as $w = (2, 5, 10, 20) \times \langle dt \rangle$ in what follows. As a result, one realization of the event dataset having $N$ event instances of $D = 5$ event types is converted into a 5-dimensional multivariate time-series of $N$ time points.

In our experimental evaluation, we used the implementation of the original authors [3][3]. For `cLSTM`, we set the number of hidden units `N_hidden = 100`, following their set-up [3]. We also set the context length `K = 10`, the $\ell_2$ regularization strength `lam_ridge = 0.01`, and the learning rate `lr = 0.001`. The group lasso strength was chosen from 14 values one by one: `lam` $\in \{0.1, 0.2, 0.3, 0.4, 0.5, 0.6, 0.7, 0.8, 0.9, 1, 2, 4, 6, 8\}$. For `cMLP`, we used `lag = 10` and chose the (hierarchical) group lasso strength from 12 values one by one: `lam` $\in \{0.01, 0.02, 0.03, 0.05, 0.1, 0.2, 0.4, 0.6, 1, 1.5, 2, 3\}$. All the other parameters, `N_hidden, lam_ridge, lr`, are the same as `cLSTM`. For both methods, we trained the model with the `train_model_ista` function.

### B.4 Existing 'sparse' Hawkes models

We compared $L_0$`Hawkes` with two existing sparse Hawkes models. One is $\ell_1$-regularizatoin based [5] ('L$_1$') and the other $\ell_{2,1}$-regularization based [4] ('L$_{2,1}$').

The objective function of `L`$_1$ is given by

$$\sum_{k,l=1}^{D} (Q_{k,l} \ln A_{k,l} - H_{k,l}A_{k,l}) - \frac{\nu_A}{2} \|A\|_F^2 - \tau \sum_{k=1}^{D} \sum_{l=1}^{D} |A_{k,l}| \tag{B.15}$$

where $\|A\|_F^2 = \sum_{k,l=1}^{D} A_{k,l}^2$. Because the domain of the solution is $A_{k,l} > 0$, the $\ell_1$ term is viewed as another linear term. The solution can be analytically given by

$$A_{k,l}^* = \frac{1}{2\nu} \left\{ -(\tau + H_{k,l}) + \sqrt{(\tau + H_{k,l})^2 + 4\nu Q_{k,l}} \right\} \tag{B.16}$$

---

[2] `https://scikit-learn.org/stable/modules/generated/sklearn.gaussian_process.GaussianProcessRegressor.html`

[3] `https://github.com/iancovert/Neural-GC`

On the other hand, the objective function of $L_{2,1}$ is given by

$$\sum_{k,l=1}^{D} (Q_{k,l} \ln A_{k,l} - H_{k,l} A_{k,l}) - \frac{\nu_A}{2} \|A\|_{\mathrm{F}}^2 - \tau \sum_{k=1}^{D} \sqrt{\sum_{l=1}^{D} A_{k,l}^2}, \qquad (B.17)$$

which no longer has an analytic solution. By equating the derivative to zero, we have

$$A_{k,l} = \frac{-H_{k.l} + \sqrt{H_{k,l}^2 + 4Q_{k,l}(\nu + \frac{\tau}{r_k})}}{2(\nu + \frac{\tau}{r_k})}, \quad \text{where} \quad r_k = \sqrt{\sum_{l'=1}^{D} A_{k,l'}^2}. \qquad (B.18)$$

These equations are iteratively computed until convergence to get a solution $A^*$.

As discussed in the main text, one immediate observation here is that these optimization problems have a singularity at $A_{k,l} = 0$. As Theorem 1 in the main text states, this prohibits $A_{k,l} = 0$ from being a solution.

In the main text we reported on the result with $\nu_A = 10^{-9}$ not to obfuscate the solution and to focus on the ability of sparsification. We have confirmed that the result is insensitive to $\nu_A$ in this `Dense10` dataset; The resulting solution always has a smooth profile in contrast to that of the $\ell_0$-constrained solution.