# OpenReview forum: "Cardinality-Regularized Hawkes-Granger Model"
_NeurIPS.cc/2021/Conference — NeurIPS 2021 Poster_

### Official Review · Reviewer_qZ8d · 2021-07-12

**Rating:** 6
**Confidence:** 2

**Summary:**

The authors a sparse Granger-causal learning framework for temporal event data following Hawkes process. The method utilizes the L0-regularized minorization-maximization (MM) learning framework, which remedies the issues of singularity in existing approaches.  The authors validated the proposed framework with two real use-cases, one from the power grid and the other from the cloud data center management domain.

The contributions of this paper seem to be as follows: (1) The proposed method is the ﬁrst L0-regularized MM learning framework for the Hawkes process applied to simultaneous instance- and type-level causal analysis. (2) The proposed method can explicitly perform simultaneous instance- and type-level causal analysis.


**Ethics Review Area:**

["I don’t know"]

**Limitations And Societal Impact:**

The limitations were mentioned. The societal impact was not mentioned but the reason was described.

**Main Review:**


There are some unclear points in this paper as described below. Although the proposed method has some novelty and the experimental results demonstrate that the performances of the proposed method were better than those of baselines in type-level tasks, due to the lack of clarity, I cannot provide higher ratings in this paper.

Specific comments:
1. L132 “When D > 1, the summation over n cannot be performed in the second term due to d_n being dependent on n. “ I cannot understand why the first term can be computed but the second term cannot be computed when D > 1 because the first term also uses d_n.
2. Theorem 1 has no proof and may be better to describe as a definition (seems to be obvious for me).
3. L252 cLSTM and cMLP are not state-of-the-art Granger causal learning algorithms. For example, economic statistical recurrent unit (eSRU; Khanna & Tan, 2019, ICLR) and generalized VAR (Marcinkevics &  Vogt, 2020, ICLR) are more efficient and effective Granger causal learning algorithms.
4. L268 GBR -> GPR?
5. Although the authors mentioned that “Power grid failure diagnosis” is a type-level causal diagnosis task and “data center warning event diagnosis” is an instance-level task, are the first two experiments type-level tasks? If yes, the authors demonstrate the performances of the proposed method were better than those of baselines in type-level tasks, but they are unknown in an instance-level task because there is no comparison with baselines in “data center warning event diagnosis” task. In the appendix, the authors mentioned a hard-windowing approach, but just mentioning and no numerical results. The authors introduced other approaches in the related work section, thus the comparison with baselines in an instance-level task may be required.

**Time Spent Reviewing:**

7 hours

---

> ### Author Response · Authors · 2021-08-08
> **Replies to the specific comments**
>
> Thank you for correctly summarizing our paper. We are sorry that the paper has a typo (yes, GBR should be GPR), and some parts of the paper confused you a bit. If allowed, we will definitely improve the readability of the text in the final version, based on your comments. Here are our answers to your specific questions/comments.
>
> **"Why can't we carry out the summation of the second term of Eq (6) over n?"**
>
> We should have been more specific. It means that "the integrals cannot be merged into a single integral by summing over n." For example, for the $\mu_d$ term (constant) in $\lambda_{d_n}$, what we meant is as simple as
> \begin{align*}
> \sum_{n=1}^N\int_{t_{n-1}}^{t_n}\mathrm{d} u \mbox{ }  \mu_{d_n}  = \sum_{d=1}^D \mu_d \sum_{n=1}^N \delta(d,d_n) (t_n -t_{n-1} )  \neq  \sum_{d=1}^D \mu_d  \times (t_N-t_0),
> \end{align*}
> where $\delta(d,d_n)=1$ if $d=d_n$ and 0 otherwise.
>
>
> **"Theorem 1 should be the definition"**
>
> We see your point, but in the literature, this is one of the common ways of associating Granger causality with the Hawkes process. See [44; Xu+16] for such an example.
>
>
> **"Other neural causal learning methods such as eSRU and generalized VAR should be compared."**
>
> We appreciate your specific comment. Our evaluation covers two main neural Granger learning frameworks, which eSRU and gVAR in fact belong to. As we are not necessarily interested in comparing specific choices of neural network design (Lines 159-171), we believe that the coverage of the evaluation is scientifically reasonable. Further technical detail is as follows.
>
> There are mainly two types of neural Granger learning frameworks known to date:
>
> - Those based on the vector autoregressive model
> - Those based on the state-space model (referred to also as “RNN-type model” in LINE 275)
>
> The first category includes cMLP and gVAR, while the second category includes cLSTM and eSRU. In each category, different methods use slightly different parameterization, but the overall architecture is the same.
>
> The critical point here is that both methods implicitly assume equi-time-interval input, as mentioned in Lines 66 and 274-275. Our intention is to empirically demonstrate that both frameworks are doomed to fail in analyzing point process data, regardless of the detail of neural network design, as long as they assume equi-time-interval input. We will make this point clearer in the final version if allowed.
>
>
> **"There is no comparison with baselines in the instance-level causal discovery task in Fig. 6"**
>
> We understand your concern, but we may not be able to define a meaningful comparison scenario here. Here are the main reasons:
>
> 1. To the best of our knowledge, the (linear) Hawkes process with the MM formulation is the only viable instance-level causal discovery method without the equi-interval assumption (see Line 66 and the comment below).  Therefore, the comparison will necessarily be a self-comparison against the other MM models.
>
> 2. Our goal here is to provide a real-world narrative on how guaranteed sparsity helps instance-level causal diagnosis (Lines 285-286). We have already mathematically and empirically shown that the baseline MM approaches do not provide sparsity (Theorem 2, Figs. 4 and 5).  Since instance-level sparsity is a direct consequence of the class-level sparsity of the impact matrix (Lines 173-174), we would have to repeat the same discussion as Figs.4 and 5, which should be redundant.
>
> *Detailed comment on potential baselines*: We are aware of a few other methods in the literature that seemingly look applicable to instance-level event causal diagnosis, but in reality, it is NOT the case. As mentioned in Lines 67-69, attention-based neural architectures typically assume equi-time-interval input. Newer transformer-based architectures also inherit the idea of equi-time-interval since they typically use sinusoidal functions to encode time.

---

> > ### Comment · Reviewer_qZ8d · 2021-08-31
> > **Thank you for the response**
> >
> > Thank you for the response.
> > I almost seem to understand your responses.
> > However, it may be difficult to read such things in the current paper. I hope the related presentation will be improved.

---

> > > ### Author Response · Authors · 2021-09-01
> > > **Thank you for taking time**
> > >
> > > Thank you for taking time to read our response. If allowed, we will definitely update the text to better describe the points you raised.

---

### Official Review · Reviewer_YomF · 2021-07-14

**Rating:** 6
**Confidence:** 3

**Summary:**

This work introduces a l_0-regularized multivariate Hawkes process model for learning sparse Granger causality from temporal event data.
On the one hand, the authors show that the prior related work cannot infer the true sparsity using l_1 or l_{1,2} regularizers.
On the other aspect, most neural point processes for learning Granger causality, assume the equi-time-input, which prevents them from analyzing continuously observed events.
The authors used synthetic data to compare their introduced one with neural point processes, and the counterpart model using l_1 and l_{1,2} regularizers, and demonstrate the new method can infer the true sparse impact matrix behind the events.

**Ethics Review Area:**

["I don’t know"]

**Limitations And Societal Impact:**

-The authors do discuss some limitations of the regularized Hawkes processes in inferring Granger causal graphs from temporal event data.
- Overall, the contribution of the paper is useful but seems to be incremental.

**Main Review:**

-Originality
To me, the new contribution by introducing l_0 regularizer, seems to be still incremental although it is the first attempt to exploit the l_0 regularizer in temporal point processes, and the other type of regularizers cannot infer the true sparse Granger causal graphs.

-Quality
The technical aspect of the paper appears to be sound. In particular, the new finding about the limitation of the l_1 and l_{1,2}-regularized Hawkes processes (HPs) in estimating true sparse impact matrices, is useful for related practitioners.

-Clarity
The paper is well written. Specifically, the limitations of the prior l_1, l_{1,2} -regularized HPs in learning true sparse Granger causal graphs, are well explained and illustrated.

-Significance
To me, the novel l_0-regularized HP , and the proof of the limitations of the prior HPs in learning true sparse Granger causal graphs, are useful and important. Nonetheless, the aforementioned contributions seem to be still insufficient to reach acceptance level.




**Time Spent Reviewing:**

5 hours

---

> ### Author Response · Authors · 2021-08-08
> **Being a rare successful example of the L0-constrained problem**
>
> Thank you for confirming our contribution of being the first work of truly sparse causal learning algorithm in the Hawkes process. If allowed, we will update the text to highlight technical hurdles of $\ell_0$ regularization problems more appropriately in the final version.
>
> As Lines 39-40 briefly mention, $\ell_0$-regularization problems are generally hard to solve. Even checking the feasibility is known to be NP-complete. **Only a handful of algorithms are known that successfully found a practical non-brute-force solution when the cardinality is regularized**. For general background of $\ell_0$-constrained optimization, the following paper may be helpful:
>
> > Christian Kanzow, Andreas B. Raharja & Alexandra Schwartz, “Sequential optimality conditions for cardinality-constrained optimization problems with applications,” Computational Optimization and Applications, (2021) 80:1, 185-211.
>
> Because of the grave difficulty, people typically solve a relaxed version of the problem with $\ell_1$ or $\ell_{2,1}$ regularization. What we have shown for the first time is that such a standard strategy fails despite the fact that the $\ell_1$ and $\ell_{2,1}$ strategies have been believed to provide valid sparsity for years. Our work presents **not only a remarkable discovery of the failure but also one of rare successful examples of the L0-regularized optimization problem.** If debunking a years-long misconception and providing a rare solution were incremental, we wonder what else could be non-incremental. With all due respect, we would like you to take a second look at the significance of our contributions.

---

> > ### Comment · Reviewer_YomF · 2021-08-24
> > **Update**
> >
> > I have read the authors' feedback and the other reviewers' comments. Regarding the authors' comments "truly sparse causal learning algorithm in the Hawkes process", i think it is necessary to tell the difference between Granger causality and causality, and appropriately position your work. I decide to update my score.

---

> > > ### Author Response · Authors · 2021-08-24
> > > **Appreciated**
> > >
> > > Thank you very much for kindly updating the score. We totally agree that the current version needs further clarifications on the very notion of causality as well as general technical challenges of the $\ell_0$ regularization problem.  If allowed, we will definitely update the paper exactly as you pointed out. Thank you again for bringing up those essential issues.

---

### Official Review · Reviewer_NuTX · 2021-07-25

**Rating:** 6
**Confidence:** 4

**Summary:**

The paper presents a L0-regularized algorithm to learn sparse Hawkes-Granger models. It is well motivated from the numerical instability issue of general MLE based approach to learn MHP, and the singularity issue of existing MM-based approach that may prohibit any sparse solution. The paper is clearly presented, and empirical results from both synthetic and real datasets show competitive performance compared to other SOTA methods in learning sparse hawkes-granger models.

**Limitations And Societal Impact:**

No potential negative societal impacts are involved in this paper.

**Main Review:**

This paper proposes an interesting L0-regularized algorithm to learn sparse Hawkes-Granger models. It is well motivated from the numerical instability issue of general MLE based approach to learn MHP, and the singularity issue of existing MM-based approach that may prohibit any sparse solution. It has a nice analysis of why the existing L1-regularization methods may fail to recover the sparse solution. Then, it builds a connection between the Minorization-Maximization (MM) and the \epsilon-sparsity technique used for probabilistic mixture models. One additional benefit of introducing the auxiliary q_{n,i} into the optimization is that they are able to capture the instance level influence per past temporal event. Experiments on both synthetic and real datasets show that the new method is able to better recover sparse structures compared to other SOTA methods.

Limits:

The root cause of the singularity issue lies in the fact that the intensity function has to be strictly positive. The base positive background intensity for a Hawkes process still guarantees this condition when no event happens at atll. The \epsilon-sparsity technique trades off this issue by introducing the additional thresholding hyperparameter \epsilon. For the existing SOTA methods, the recovered parameters of the mutual excitation matrix can also be thresholded using a similar thresholding hyperparameter \epsilon which can be tuned using the dev dataset. I wonder how this can be compared to the proposed method. In addition, for the synthetic experiment, it will be more convincing to report the F1 metric by comparing the recovered sparsity structure to the groundtrue structure rather than showing qualitatively the sparse pattern in Figure 4. Finally, it will be more convincing to add event time prediction using MAE to evaluate the predictive performance of the new method against the baselines.


**Time Spent Reviewing:**

2-2.5 hours

---

> ### Author Response · Authors · 2021-08-08
> **Reply to the three main questions/comments**
>
> We appreciate your comments. You have mainly three comments:
>
> **"What's wrong with simply putting a threshold on the impact matrix to get a sparse causal pattern?"**
>
> We see your point, but we respectfully remind you that the proposed $\ell_0$ regularization method is NOT just thresholding. As mentioned in Lines 283-284 in Section 6, it gives a qualitatively different solution from "simple thresholding by the magnitude."
>
> The beauty of our $\ell_0$ regularization approach is that the sparsity pattern is guaranteed to be optimal in the maximum likelihood sense. In other words,  your model is guaranteed to be the best fit under the strict definition of Granger causality.
>
> Arbitrarily thresholded solutions, on the other hand, do not have optimality in general. This means that (1) your model may not be a faithful representation of the data, and (2) it may even contradict the very notion of the Granger causality.
>
>
> **"Figure 4 should be quantified with the F1-score"**
>
> We totally understand your wish for clearer numbers, which we wished, too. There are two reasons why the F1 score is not a meaningful metric here.
>
> 1. The task is not binary classification in this case. Figure 4 is to show how "zeros" emerge in the impact matrix as the regularization strength increases. There is no such thing as a well-defined label.
>
> 2. Even if there were a binary label defined somehow, the $\ell_1$ and $\ell_{2,1}$ approaches cannot produce any zero as Theorem 2 shows. So, technically, the accuracy will be zero, which is not meaningful information.
>
>
> **"The performance should be evaluated in the future prediction setting"**
>
> It is absolutely true that most of the recent point process papers address a prediction task, and thus, we understand why you are suggesting "event time prediction using MAE". However, we are afraid that such a metric would not be scientifically relevant to our task.
>
> Our goal is to learn Granger causal patterns from point process data (Section 3.1), which is an unsupervised task (Line 108). Even if a model has a good prediction performance, it may not be able to provide meaningful information on causal dependencies, as we briefly discussed in Section 4.1. In fact, we do not think that such a metric is standard in the literature when Granger causal learning (=*interpretability*) is the main goal.

---

### Decision · Program_Chairs · 2021-09-27

**Decision:**

Accept (Poster)

**Comment:**

The paper addresses an important problem, reviewers consider the method sound. There were some concerns that the work is too incremental I would not over estimate this issue given the relevance of the problem.